# Immunogenicity of RSV Fusion Protein Adsorbed to Non-Pathogenic *Bacillus subtilis* Spores: Implications for Mucosal Vaccine Delivery in Nonclinical Animal Models

**DOI:** 10.3390/biomedicines13051112

**Published:** 2025-05-03

**Authors:** Jianying Xiao, Hao Wang, Cheryl Callahan, Gregory O’Donnell, Silveria Rodriguez, Ryan P. Staupe, Carl J. Balibar, Michael P. Citron

**Affiliations:** 1Infectious Disease and Vaccines, Merck & Co., Inc., Rahway, NJ 07065, USA; jianying_xiao@merck.com (J.X.); hao_wang@merck.com (H.W.); cheryl_callahan@merck.com (C.C.);; 2Quantitative Biosciences, Merck & Co., Inc., Rahway, NJ 07065, USA; gregory_odonnell@merck.com (G.O.); silveria_rodriguez@merck.com (S.R.)

**Keywords:** mucosal vaccines, intranasal vaccines, adjuvants, *Bacillus subtilis* spores, mice, cotton rats, respiratory syncytial virus (RSV)

## Abstract

**Background/Objectives**: Mucosal vaccines are rare but commercially desirable because of their real and theoretical biological advantages. Spores and vegetative forms from Bacillus have been used as probiotics due to their stability under various environmental conditions, including heat, gastric acidity, and moisture. Preclinical studies have shown that *Bacillus subtilis* (*B. subtilis*) spores can serve as effective mucosal adjuvants. Our study aimed to evaluate B. subtilis spores as a mucosal adjuvant. **Methods and Results**: We demonstrate in rodents that the fusion protein (F) from respiratory syncytial virus (RSV), when combined with either heat-inactivated or live *B. subtilis* spores, elicits robust IgG binding and neutralizes antibody titers following both systemic and intranasal administration in mice. The spores facilitate TH-1 and local IgA responses, which could enhance antiviral protection. However, this vaccine failed to elicit measurable antibodies when immunized using a strict intranasal administration method in cotton rats. **Conclusions**: Our findings illustrate the differing immune responses between the two rodent species, highlighting the need for the careful consideration of validated methods when evaluating intranasal vaccines in preclinical studies.

## 1. Introduction

Currently, licensed vaccines against respiratory syncytial virus (RSV) exploit a pre-fusion-stabilized conformation of the fusion (F) glycoprotein from the virus. These vaccines are all administered parenterally and are approved for select populations. However, there is continued room for improvement [1]. We and others have demonstrated that preclinically mucosal vaccine candidates, using alternative routes of administration such as intranasal or oral delivery, can elicit both humoral and cellular immune responses, provide protection against infection, and do not induce airway hyperreactivity [2,3,4,5,6]. Mucosal adjuvants represent an attractive strategy for developing effective vaccines for RSV via non-systemic routes of administration [7]. *B. subtilis* spores offer one such adjuvant with a favorable safety profile that has been evaluated in both preclinical and clinical settings [8,9].

*B. subtilis* are gram-positive, aerobic, rod-shaped bacteria that are able to produce spores that are stable, heat-resistant, and have a safe profile as demonstrated by their worldwide acceptance as a probiotic [10]. Genetically engineered spores can express heterologous antigens [11,12]. These spores have been shown to stimulate systemic, local, and cellular immune responses, providing protection against infection when administered via the mucosal route [13,14,15,16,17]. Thus, Bacillus spores present an attractive vaccine delivery platform for exploring mucosal immunization.

To evaluate the full potential of this platform, we started by combining live and killed spores to the fusion glycoprotein (F) from RSV in its pre-fusion conformation (Pre F) and measured both local and systemic immune responses first in mice and later in cotton rats. The cotton rat has been used for many decades to investigate both innate and adaptive immune responses, as well as the safety and efficacy of a variety of antiviral and antibacterial prophylactics and therapeutics. Additionally, this species is valuable for studying the disease pathogenesis of a wide range of human pathogens, including viruses from the *Orthomyxoviridae* family, like influenza, as well as the *pneumoviruses*, *paramyxoviruses*, and *Staphylococcus aureus* [18,19,20,21,22,23,24,25,26,27,28]. We recently established a model for strict intranasal immunization using this species, which is semi-permissive to many respiratory infectious pathogens [29]. Our results clearly demonstrate how inadequate in vivo methods can result in misleading outcomes of immunogenicity and protection against infection. Our data emphasize the importance of robust and translatable animal models for vaccine evaluation.

## 2. Materials and Methods

### 2.1. Animal Ethics Statement

All experiments involving laboratory animals were approved by the Institutional Animal Care and Use Committee (IACUC) at Merck & Co., Inc., Rahway, NJ, USA, and conducted in accordance with the Guide for the Care and Use of Laboratory Animals. Additionally, the studies adhered to the ARRIVE guidelines of the National Center for the Replacement, Refinement, and Reduction of Animals in Research [30]. Sample size was determined based on the minimum number of animals required, aligning with the principles of the three Rs: Replacement, Reduction, and Refinement. In order to detect meaningful differences between various formulations, doses, and routes of administration, we relied on both historical data of the comparator vaccine and published reports using the innovator adjuvant (*B. subtilis*). Lastly, we did not exclude any animals or experimental data points. Animals were randomly assigned to each study group without predetermined criteria. Each sample was run simultaneously, regardless of treatment, to ensure consistency. Analysts conducting ex vivo assays were aware of the specific group assignments to accurately track each sample data point but were unaware of the objectives for each group.

### 2.2. Housing and Husbandry

All animals were housed in microisolator cages in the animal facility at Merck & Co., Inc., West Point, PA, USA. Laboratory Autoclavable Rodent Diet 5010 (LabDiet^®^ St. Louis, MO, USA) and autoclaved potable water were provided ad libitum. Animals were housed in rooms with a 6 am–6 pm light cycle and maintained at 68–77 °F with a midpoint of 72 °F with 30–70% humidity. Mice were housed in groups of six animals per cage using Bed-o’Cobs^®^ bedding and provided paper towels and nestlets. Cage floor size was 143 square inches. Cotton rats were housed in large colony boxes with Bed-o’Cobs^®^ bedding. Cotton rats were provided nestlets and at times Nylabone^®^ and or tubing for enrichment. Their cage floor was 153 square inches. Both mice and cotton rat vendors have the microbiological status of their respective colonies available via the vendor website and are routinely screened at various frequencies. Cotton rats were screened for parasites, bacteria, mycoplasma, fungi, viruses, and pathological lesions for cotton rats prior to arrival at the closed vivarium. Surveillance of the health status of the barrier production of the mice was monitored by the vendor and free of an extensive list of viruses and other pathogens using VAF/Plus^®^.

### 2.3. Mouse Immunizations

Female BALB/c mice were purchased from Charles River Laboratories (Wilmington, MA, USA). Groups of six mice for each group aged 6–7 weeks were immunized either intramuscularly (IM), intranasally (IN), or per os (PO) three times at weeks 0, 3, and 9. For the IN route of immunization, the vaccines contained 2 micrograms (µg) of RSV Pre F protein unadjuvanted, live or killed (iSpores) *B. subtilis* spores at a concentration of 1 × 10^8^ CFUs per dose, 25 µg of aluminum phosphate wet gel suspension (AdjuPhos^®^ adjuvant, (Invivo Gen San Diego, CA, USA), a combination of iSpore with aluminum or live spores with aluminum. For the IM and PO routes of immunization, the vaccines contained 2 micrograms (µg) of RSV Pre F protein with 40 µg of AdjuPhos^®^ and live spores or killed spores. See Table 1 for groupings. Whole blood for serum was collected from the tail vein at weeks 2, 5, and 11. Two weeks after the third immunization serum, spleens, and nasal washes were harvested following sacrifice using inhaled carbon dioxide (CO_2_) at 30–70%/min. verified with lack of heartbeat, respiration and movement. Serum from whole blood and nasal washes was processed and stored at 2–8 degrees Celsius (C) until analysis. Spleens were processed for splenocytes and assayed fresh.

### 2.4. Cotton Rat Immunogenicity and Efficacy Study

Four to eight-week-old female cotton rats (*Sigmodon hispidus*, Inotiv, Inc., Lafayette, IN, USA) in groups of four were immunized with various vaccine formulations. First, 2 µg of Pre F protein was mixed with either 1 × 10^8^ CFU live or killed *B. subtilis* spores (iSpores) or AdjuPhos^®^. The groups with AdjuPhos^®^ contained 25 µg per dose when administered with IN and 40 µg for intramuscular immunization (IM). Additionally, inactivated RSV, generated by heat andgamma irradiation and verified by the lack of plaques in a live virus assay was formulated with or without 1 × 10^8^ CFU iSpores. The inactivated RSV was administered at an equivalent of 0.2 µg F protein. Moreover, a single immunization of 10^5.5^ PFU RSV A2 was dosed IN at time 0 (T = 0). In general, three immunizations were given at weeks 0, 3, and 5. However, we also evaluated one, two, and three immunizations using the 2 µg Pre F with iSpores formulation. Additionally, RSV A2 was administered once. Vaccines were administered at 100 µL in alert animals for intramuscular immunization (IM) or 10 µL intranasally (IN) in sedated animals, as described previously [29]. Two weeks after the third immunization, animals were infected with RSV A2 at 10^5.5^ PFU, and lung and nose tissue was collected and homogenized four days post-infection. Whole blood was collected and processed for serum (see Table 2).

### 2.5. Vaccine Preparation of Protein Absobed with Spores

Highly purified endospores from *B. subtilis* Cohn (ATCC 23857, strain 168) were prepared using a previously described method [31,32,33]. *B. subtilis* was cultivated in Difco Sporulation Medium (DSM) (for 1 L: Bacto nutrient broth (Difco) (8 g); KCl (1 g); MgSO_4_·7H_2_O. (0.12 g); 1M NaOH (~1.5 mL); 10 mM MnCl_2_ (1 mL); 1 M Ca(NO_3_)_2_ (1 mL); 1 mM FeSO_4_ (1 mL); and H_2_O) at 37 °C for 72 h. Next, the bacterial culture was centrifuged at 10,000× *g* for 10 min. The supernatant was discarded and the pellet containing the endospores was suspended in lysis buffer (50 mM Tris-HCl, 50 μg/mL lysozyme) and incubated at 37 °C for 1 h. The spores, either live or inactivated by autoclaving at 121 °C and 15 pounds per square inch (p.s.i.) for 30 min, were washed with distilled water and used for absorption with Pre F. The protein was codon-optimized for mammalian usage, cloned into an expression vector, and transiently transfected into Expi293 suspension cells (Life Technologies Carlsbad, CA, USA). Cell culture supernatants were harvested on days 3 to 7 post-plasmid transfection and evaluated with both Western blotting and ELISA. To obtain purified Pre F, cell culture supernatants were purified using a modified method based on the previously described procedure [29]. Briefly, His-tagged proteins were purified using Ni-Sepharose chromatography (GE Healthcare Chicago, IL, USA), and tags were removed by overnight digestion with thrombin. Digestion was performed during dialysis to reduce imidazole concentration. To remove co-eluting contaminants and uncleaved F protein, samples were subjected to a second Ni-Sepharose chromatography step. F proteins were further purified by gel filtration chromatography (Superdex 200, GE Healthcare) and stored in a buffer consisting of 50 mM HEPES and 300 mM NaCl at a pH of 7.5. Finally, protein was absorbed in an Aluminum phosphate gel adjuvant with killed or live spores by simply mixing them together in PBS buffer without Ca^2+^ and Mg^2+^ with gentle rocking for 30 min.

### 2.6. Cells and Virus

Host HEp-2 cells (ATCC) were propagated twice per week in Minimum Essential Media Eagle (MEME) (Bio Whittaker Walkersville, MD, USA) supplemented with 2 mM L-glutamine, 10% fetal bovine serum (FBS, HyClone Logan, UT, USA), and antibiotics (streptomycin, 100 mg/mL; penicillin, 100 U/mL). Virus stocks Respiratory syncytial virus (RSV) strain A2 (ATCC VR-1540) and Long (ATCCVR-26 ™) were generated by infecting HEp-2 cells (ATCC, CCL-23).

A modification of the previously described trans-tracheal technique [34] was utilized to obtain nasal airway lavage (NAL) samples. The upper airway, including the palatopharyngeal region, was isolated using a dissection scissor to isolate the larynx. A 24 G IV catheter was inserted through the pharyngeal opening and 300 µL of saline was instilled and draining fluid was collected from the nostrils using a microtube. NAL was clarified using centrifugation and the resulting supernatants were stored at −70 °C. Graphs and analysis were performed using GraphPad Prism version 10.2.2 for Windows, GraphPad Software, Boston, MA, USA, www.graphpad.com.

### 2.7. Determination of Serum IgG Titers and IgA Titers in NLF Samples in Mouse and Cotton Rat

Antibody binding titers against pre-fusion RSV F were evaluated using ELISA. 384-well MaxiSorp-treated plates (Thermo Scientific Waltham, MA, USA) were coated with 2 μg/mL of purified recombinant RSV F protein and incubated overnight at 4 °C. Plates were then washed and blocked using 3% nonfat milk in 1x PBS-T (blocking buffer) for 30 min at room temperature. Animal sera was screened in a 10-point titration series, starting from an initial 50-fold dilution with subsequent 4-fold dilutions. All serum was diluted in blocking buffer. Following sample titration, the serial dilutions were added to the coated and blocked assay plates and incubated at room temperature for 2 h. Plates were then washed six times with 1x PBS-T. Following the washing step, HRP-conjugated anti-species secondary antibodies were added: goat anti-mouse antibody (Thermo Scientific, mouse total IgG) diluted 1:10,000 in blocking buffer; 1:8000 dilution in blocking buffer of rat anti-mouse IgG1 or IgG2a (Abcam, mouse IgG1 or IgG2a assays); and 1:2000 dilution in blocking buffer of chicken anti-cotton rat (ICL lab, cotton rat total IgG). The plates were incubated at room temperature for 1 h. Plates were washed again six times with 1x PBS-T and developed with West Pico PLUS Chemiluminescent Substrate (Thermo Scientific). After a 15 min incubation at room temperature, luminescence was read on the EnVision 2104 microplate reader (PerkinElmer Waltham, MA, USA). An interpolated endpoint titer was calculated for each serum sample using the relative light unit (RLU) values and the following formula: interpolated endpoint titer = (starting dilution/series dilution factor) X (series dilution factor^t) where t = x − [(cut-off − L)/(H − L)]. The “cut-off” value was designated as 50,000. “H” = High well RLU value (the RLU value of the first titration point ABOVE 50,000), and “L” = Low well RLU value (the RLU of the first titration point BELOW 50,000). x = Low well number (the number in the titration series of “L”, where the first dilution in the titration series was 1, and the highest serum dilution of the titration series was 10). Samples that did not cross the cut-off value were given a placeholder titer of one-half the initial starting dilution.

The measurement of IgA titers in nasal lavage fluid (NLF) with ELISA was assayed in a comparable manner to serum samples, with the biggest difference occurring in the preparations of the sample dilutions. For NLF samples, an initial dilution of 3-fold was made using blocking buffer, followed by 10-fold titration. Once the NLF titrations were prepared, the RSV F protein-binding ELISA for IgA antibodies was carried out as described above, with the only difference being the use of the rat anti-mouse IgA-HRP-conjugated secondary antibody (Southern Biotech, diluted 1:15,000 in blocking buffer) used for IgA detection. The interpolated endpoint titers were determined in the same manner as described above for serum IgG titers. Graphs and analysis were performed using GraphPad Prism version 10.2.2 for Windows, GraphPad Software, Boston, MA, USA, www.graphpad.com.

### 2.8. Serum Neutralization Assays (SNA)

The serum neutralization assay for determining the functional capacity of serum antibodies to block virus entry into the cell was performed as previously described [35]. Briefly, sera were heat-inactivated and serially diluted in a 384-well plate and combined with the virus for 1 h. Cells were added at a density of 5000 cells per well and cultured for 72–96 h at 37 °C. Following this, media were revolved and contents were lysed using AlphaLISA lysis buffer (PerkinElmer) for 60 min at room temperature. Diluted lysates from a set of four 384-well plates were combined into a 1536-well plate and exposed to a suspension of AlphaLISA acceptor beads (PerkinElmer) and biotinylated antibody for 1 h at room temperature. A suspension of streptavidin-coated donor beads (PerkinElmer) was added to the plate and incubated at room temperature for 30 min, after which fluorescence was measured on an EnVision microplate reader (PerkinElmer). Four-parameter curve fitting BioAssay analysis software v12.11.0.0 (Hayward, CA, USA) was used to calculate titers, from which the 50% neutralizing titer (NT50) was derived at the curve inflection point and reported as fold dilution. Graphs and analysis were performed using GraphPad Prism version 10.2.2 for Windows, GraphPad Software, Boston, MA, USA, www.graphpad.com.

### 2.9. Intracellular Cytokine Staining (ICS)

Mouse splenocytes were cultured in R10 medium (RPMI 1640 supplemented with 10% of fetal calf serum, 1 × penicillin/streptomycin, 1 mM MEM sodium pyruvate, 2 mM L-glutamine, 10 mM 4-(2-Hydroxyethyl)-1-piperazine ethane sulfonic (HEPES) acid buffer, and 50 nM 2-Mercaptoethanol) at 37 °C in a 4–6% carbon dioxide incubator in round-bottomed plates. A 2 μg/mL concentration of RSV F peptide pools (15-mers of the entire F sequence overlapping by 11-mers, custom order JPT, Germany) and anti-mouse CD28 (clone 37.51) and CD49d (R1-2) costimulatory antibodies (BD Biosciences Franklin Lakes, NJ, USA) was added into each assay well at final concentrations of 2 μg/mL. The plates were incubated at 37 °C for 60 min. After incubation, freshly diluted brefeldin A was added to each sample at 10 micrograms per milliliter final concentration, and the plates were incubated for an additional 5 h. After 6 total hours of stimulation, 20 uL of 20 mM EDTA in PBS was added to each well, and plates were stored at 4 °C overnight. To detect cytokine-producing T cells, samples were washed once with PBS and stained with LIVE/DEAD™ Fixable Violet (Invitrogen Carlsbad, CA, USA) for 15 min at room temperature. Samples were washed with FACS buffer (PBS with 1% FBS and 0.01% sodium azide) and stained with a cocktail of fluorescently labeled antibodies specific to cell surface markers CD3 (clone 145-2C11), CD4 (RM4-5), and CD8 (53-6.7) (BD Biosciences). After 30 min incubation at 4 °C, cells were washed with FACS buffer and incubated with 200 µL/well BD cytofix solution (BD Biosciences) at 4 °C for 25 min, washed twice with BD perm wash buffer (BD Biosciences), and stained with a cocktail of fluorescently labeled anti-cytokine antibodies (TNF-α clone MP6-XT22, IFN-γ clone XMG1.2, and IL-2 clone JES6-5H4, all from BD Biosciences) for 35 min at 4 °C. The cells were washed twice with BD perm wash buffer and resuspended in 200 µL BD stabilizing fixative. The fluorescent signals were analyzed using an X-50 flow cytometer (BD Biosciences) and analyzed in OMIQ (Dotmatics Boston, MA, USA). Graphs were performed using GraphPad Prism version 10.2.2 for Windows, GraphPad Software, Boston, MA, USA, www.graphpad.com.

### 2.10. RSV Viral Titers in Lungs and Nose

Lung and nose viral titers were determined using a plaque assay on HEp-2 cells. HEp-2 cells were pre-seeded and incubated at 37 °C on the day before the assay. Briefly, homogenate tissue samples were diluted and added in duplicate to either a 24-well or 96-well plate containing confluent cell monolayers. The liquid was aspirated after incubation for 1 h at 37 °C followed by the addition of 0.75% methylcellulose. After a five-day incubation at 37 °C, cells were fixed and stained with crystal violet/glutaraldehyde solution. Viral plaques were counted and titers were expressed as pfu/g of tissue. Graphs and analysis were performed using GraphPad Prism version 10.2.2 for Windows, GraphPad Software, Boston, MA, USA, www.graphpad.com.

## 3. Results

### 3.1. Antibody and T Cell Responses in Mice Immunized Intramuscularly, Intranasally, and PO

In a study aimed at determining the immunogenicity of the mucosal administration of a vaccine formulated with Pre F protein and *B. subtilis* spores, we compared systemic immunization against oral and intranasal delivery of the same vaccine formulation in Balb/c mice. Three immunizations of the vaccine were given. The vaccines contained 2 μg of protein and were administered 3 or 6 weeks apart; however, depending on the route of administration, we used various volumes. The formulations included inactivated and live spores, aluminum adjuvant, or no adjuvant, with PBS buffer without Ca^2+^/Mg^2+^. The routes of administration consisted of intramuscular (IM) at a volume of 0.1 mL, intranasal (IN) at a volume of 0.01 mL, or orally (PO) at a volume of 0.5 mL (see Figure 1). Serum IgG antibody using ELISA (see Figure 2A) shows comparable endpoint titers for all adjuvanted immunizations when 2 µg of Pre F was delivered systemically or intranasally despite the type of spore (the GMT range was 1341 to 1.38 million), eleven weeks after three vaccinations. A clear increase from the first to the second and third immunization was observed in the IN groups. A boost from the first to the second, with no clear benefit from a third immunization, was observed in the IM groups. No antibody titers above the set background were observed in the PO groups (see Figure 2A). Serum-neutralizing antibody titers (SNab) for the IM and IN groups corresponded proportionally to the binding titers. Unlike the binding titers, the SNab showed a modest boost from the second to third immunization for the IM route of administration (see Figure 2B). SNab titers after three immunizationssignificantly differedfor all IN groups, compared to the live spore IM group, except for the live spore and aluminum IN group. The group with aluminum only was close to 2-fold lower compared to when the spores were added when the vaccine was delivered intranasally (see Figure 2B).

The highest IgA responses were elicited by live spores after IN delivery in mice compared to inactivated spores and aluminum (GMT 32 vs. 13, respectively). IgA responses were not elicited by either protein alone or IM immunization, as expected. Interestingly, the group formulated with protein and aluminum only, while not often considered a potent mucosal adjuvant, did elicit appreciable (13) specific IgA titers from the nasal lavage samples (see Figure 3).

The IgG2a/IgG1 ratio for all groups is positive since IgG1 was greater (see Figure 4B). Typically, IgG2a:IgG1 ratio > 2 is indicative of a Th1-type response, an IgG2a:IgG1 ratio < 0.5 is indicative of a Th2-type response, and 0.5–2 is viewed as a Th0 or Th1/Th2-mixed response [36,37]. We observed generally higher IgG1 responses in all the groups for all the routes of immunization (see Figure 4A). The IN group IgG2a titers were 70–30,000, with the lowest titer due to lacking adjuvant. The IM group was between 10,000 and 30,000. The IgG1 was 3000 to two million for the IN group and two million for the IM group. The lack of an overwhelming favorable Th1 response is not surprising. The vaccine contained a soluble antigen with aluminum adjuvant, typically known to skew towards Th2 in rodents, even though it is generally known to be more mixed in humans. Also, it is important to emphasize that Balb/c mice were used in this study, knowing that this strain of mice areTh2-dominant. Interestingly, the ratio in the two groups lacking aluminum salts, while not significant, is slightly elevated, suggestive of a Th1 trend. The oral immunization results are not shown since all were below the lower limit of detection. CD4+ and CD8+ T helper cellular responses are detectable only in the systemically administered groups (see Figure 5).

### 3.2. Immunogenicity and Efficacy in Cotton Rats Immunized with Pre F and Spores, IM and IN

We were unable to detect antibodies against the RSV F protein when delivered orally co-formulated with *B. subtilis* in mice but were able to elicit anti-F antibodies via the intranasal (IN) route. To access the immunogenicity of F protein combined with *B. subtilis* spores, we used a validated method for strict IN immunization in cotton rats. Female cotton rats (*Sigmodon hispidus*) aged 3–7 weeks were divided into groups of four. All immunizations were conducted following strict intranasal methods. Inactivated respiratory syncytial virus (RSV), which was prepared through gamma irradiation and heat-killing techniques with or without iSpores, was administered three times at two or three-week intervals. RSV strain A2 was given once. Additionally, Pre F protein was mixed with either spores or AdjuPhos^®^ and delivered three times, adhering to the same two or three-week interval protocol. Also, a Pre F and iSpore combination was administered in varying frequencies: once, twice, or three times, also separated by two or three weeks. Lastly, the combination of the protein with iSpores or AdjuPhos^®^ was delivered via intramuscular (IM) injection Table 3.

Serum samples were analyzed for mean ELISA titers. Notably, both live and inactivated virus as well as the protein combined with spores yielded mean endpoint titers above the lower limit of detection (LOD). However, functional neutralizing antibodies were not detectable in the groups immunized with the protein alone. IM immunization with the protein in combination with either iSpores (mean binding titer: 5 × 10^6^; NT50: 5 × 10^2^) or AdjuPhos^®^ (mean binding titer: 9 × 10^6^; NT50: 1 × 10^3^) effectively elicited antibody responses (see Figure 6A,B). Antibody titers were inversely correlated with the viral load detected in both the lower and upper respiratory tracts four days post-infection with RSV A2 at a dosage of 1 × 10^5.5^ pfu. Consistent with previous observations, the live virus, showed lower immunogenicity compared to IM administered vaccines, but completely reduced the virus in the respiratory compartments Table 4.

Figure 7 Cotton Rat Immunogenicity and Efficacy Study using Strict Intranasal Immunization Methods.

### 3.3. Immunogenicity in Cotton Rats Immunized with Pre F and Spores Using Methods for Strict and Non-Strict Intranasal Immunization

To determine if strict and non-strict intranasal immunization results in a different antibody profile, we conducted a second cotton rat study comparing different intranasal immunization methods. The vaccines all contained 2 μg of Pre F protein containing 1 × 10^8^ CFU iSpores. The vaccine was administered IN, three times, three weeks apart, using 0.01 mL for strict intranasal immunization and 0.05 mL and 0.1 mL for non-strict IN immunization (see Figure 8A). Anti-RSV serum IgG with ELISA (see Figure 8B) shows comparable endpoints for both 0.05 mL and 0.1 mL volume groups. The group receiving 0.01 mL showed antibody responses that were below the lower limit of detection. Serum neutralization antibody responses showed a similar profile (see Figure 8C and summarized in Table 5).

Strict (0.01 mL volume) vs. non-strict intranasal (0.05 and 0.1 mL volume) immunization in cotton rats using 2 mcg RSV Pre F protein with 1 × 10^8^ CFU killed spores (iSpore) administered equally over both nostrils under ketamine and xylazine anesthesia. IgG-binding ELISA using RSV Pre F protein as a coating antigen and geometric titers (GMT) was determined as an interpolated endpoint and 95% confidence interval (CI) for upper and lower CI (95% CI [U, L) using serum collected two weeks after the third immunization.

## 4. Discussion

Limited adjuvants currently exist for developing mucosal vaccines. Various modified bacterial toxins such as heat-labile toxins (LT); Vibrio cholerae, cholera toxin (CT); pathogen-associated molecular pattern (PAMP) molecules such as bacterial flagellin; other bacteria-derived proteins such as outer membrane proteins (OMPs); and polymer micro and nanoparticles have been evaluated as mucosal adjuvants, preclinically [38]. *B. subtilis* spores are agents that have also been identified to possess adjuvant properties [39,40,41,42]. Bacterial spores can be modified to express heterologous antigens. Their ability to withstand the harsh and often hostile mucosal compartments offers an advantageous and safe approach as an immunization strategy [43,44,45,46]. The mucosal delivery of spores expressing heterologous antigens has theoretical benefits such as stimulating immune responses in the mucosa, the port of entry for many pathogens [47,48,49]. Additionally, ease of use and lack of needles may potentially allow for at-home administration, which may be an attractive feature for increased compliance and pandemic preparedness.

While germinating live and inactivated spores have been shown to be immunogenic under many different regimens and delivery systems, recombinant and non-recombinant approaches to utilizing spores differ. For instance, the non-recombinant approach often requires additional investment for purified proteins. Conversely, it lacks the drawback of introducing genetically modified microorganisms into the environment. To capitalize on the attractiveness provided by spores, we use purified Pre F from RSV, a highly immunogenic and efficacious glycoprotein [50,51,52] mixed with viable and inactivated *B. subtilis* spores.

We first demonstrated in mice that our vaccines could increase antibody titers when administered both systemically and intranasally. However, we failed to elicit measurable immune responses when the vaccine was delivered via oral administration. Generating antibodies following oral immunization with *B. subtilis* spores has been demonstrated previously. However, robust antibody responses were generally observed using the recombinant approach [53,54,55]. Huang et al. did demonstrate a non-recombinant approach using *C. perfringens* alpha toxin antigens that was able to elicit antibodies, although it provided significantly less protection [56]. Unlike the aforementioned study, we did not assess the quality attributes of the spores for surface hydrophobicity, charge, or absorption capacity. Additionally, our protein, the meta stable trimeric RSV Pre F, is noticeably larger than those evaluated previously evaluated. This size difference may suggest that specific spores combined with native proteins can vary in degree of adsorption and display [57].

Regardless of whether antigens are displayed during sporulation, protein anchoring or spontaneous absorption, the effective presentation of the antigen to immune cells is essential [58]. As a precursor to designing heterologous antigens expressed recombinantly on spore surfaces, we chose to first assess spores combined with our purified protein, Pre F. Recombinant approaches may be more desirable for oral immunization, but data suggesting spores mixed with antigens, such as TTFC of *C. tetani* and PA from *B. anthracis*, can induce measurable immune responses [59]. It is also known that spores alone can elicit antibodies, albeit at lower levels [60]. We included an intranasal arm in our study because it has been demonstrated that inactivated influenza virus, when combined with killed *B. subtilis* spores and delivered via the nose, had favorable outcomes in mice [61]. Furthermore, RSV naturally favors the respiratory tract over the gastrointestinal tract.

Our findings in mice indicated that RSV Pre F protein and *B. subtilis* spores did elicit humoral responses and very low levels of cellular responses in mice following IM and IN administration. We also observed mucosal IgA in the IN groups but not in the IM groups. Interestingly, we also observed the IgA antibody in the IN group with protein adjuvanted with aluminum. Generally, aluminum salts are not commonly regarded as strong mucosal adjuvants; however, preparations like Alhydrogel^®^ have been shown to produce robust antigen-specific immune responses both systemically and in mucosal secretions following intranasal immunization [62]. In our study, we used Adjuphos^®^, an aluminum phosphate wet gel suspension. Unlike Alhydrogel, the negatively charged Adjuphos^®^ is known to induce Th2 responses and dissolve more readily. Nonetheless, this serves as another example of an aluminum salt eliciting IgG and IgA from nasal samples after IN immunization. Furthermore, CD4+ T cell cytokine responses were observed with aluminum co-mixed with the protein following systemic administration, as indicated by the expression of IL-2 and TNF-α from pooled mouse splenocytes stimulated with peptides. Comparable CD8+ T cell cytokine production, including IL-2, IFN-γ, and TNF-α, was observed when using the protein plus killed spores as the vaccine.

While the immunogenicity from the *B. subtilis*-containing vaccines in mice was encouraging, we wanted to verify the method of intranasal delivery. It is crucial to differentiate between intranasal delivery and nasal vaccination in preclinical animals since the interpretation of the results may not translate across species, including in humans [63]. We administered what appeared to be a large inoculum to the mice and were concerned that we might have inadvertently immunized them intrapulmonarily, potentially leading to misleading outcomes. Both we and others have demonstrated differences in immune responses between the upper and lower respiratory tracts and validated methods for strict intranasal (IN) and intrapulmonary immunization in preclinical species [29,64]. Specifically, the type of anesthesia and volume are critical factors. Additionally, the nasal compartment contains regions of importance, such as the organized nasopharynx-associated lymphoid tissue (O-NALT) and Waderyer’s ring, which are key sites for inducing mucosal immunity [65]. These and other mucosal structures and anatomical features not only differ between humans and rodents but also among various preclinical species [66,67,68]. Beyond structural differences, other translational considerations may contribute to outcome disparities. For instance, cotton rats have an intact Mx system unlike some other rodent species [69]). Furthermore, delivery systems for nasal delivery may vary between clinical and preclinical studies. Atomized and spray systems that measure the amount and particle size of the vaccine may not accurately represent how animals in the research setting are immunized. Vaccine deposition deep within the lungs, where the antigen has a longer exposure opportunity, does not constitute true intranasal immunization and may not translate easily to humans. Therefore, we evaluated the same vaccines used in the mouse study in cotton rats with validated methods for strict IN immunization.

In the cotton rat studies, we observed an absence of antibodies and corresponding protection against infection using the same protein-containing *B. subtilis* vaccine used in our mouse study, but only when employing established strict IN methods. The lack of sufficient binding and neutralizing antibodies was inversely correlated with the presence of the virus in the respiratory tract tissues. Additionally, we verified the ability to detect both binding and functional antibodies using non-strict IN immunization methods in a second cotton rat study. These findings indirectly suggest that the immune responses observed in the intranasal (IN) groups in our mouse study may be influenced by unintentional immunization of the lower respiratory tract (LRT), which could lead to inconsistencies in immune outcomes following vaccination.

There were limitations to our investigation. We did not include a vaccine containing a well-established mucosal adjuvant such as cholera toxin B subunit (CTB) in order to limit the use of live animals. However, we do know this protein with CTB is both highly immunogenic and protective against RSV infection in the cotton rat model using strict intranasal methods. To directly assess true intranasal (IN) immunization in mice, potential strategies could be explored to refine intranasal immunization methodologies [70,71,72] or develop new methodologies. We chose not to investigate what constitutes strict IN methods in mice to minimize live animal use in establishing these models. Furthermore, to fully evaluate the utility of *B. subtilis* as a mucosal adjuvant, it could be important to characterize the absorption capacity of the proteins, incorporate proteins from enteric viruses for oral immunization, and/or engineer recombinant versions. While we do not fully understand the mechanism of absorption or how they mediate immune enhancement, we do know the importance that humoral and cellular adaptive immunity, such as T cells (CD4 and CD8) and antibody-specific IgG and IgA, has on the control of viruses such as RSV [73]. We did not interrogate the modulation or look for the recruitment of innate immune cells such as neutrophils and NK cell infiltrates, nor did we challenge the animals with the virus to evaluate viral reduction from immunization. These points would need to be addressed with subsequent experiments. Nevertheless, these findings do highlight the importance of using validated experimental in vivo methods to avoid ambiguous interpretations of the outcomes.

## 5. Conclusions

Using a vaccine composed of RSV F protein and *B. subtilis* spores, we were unable to detect antibodies in mice via the oral route contrary to intranasal and systemic immunization. However, the same vaccine failed to elicit measurable antibodies when administered intranasally in a cotton rat model validated for strict intranasal immunization.

## Figures and Tables

**Figure 1 biomedicines-13-01112-f001:**
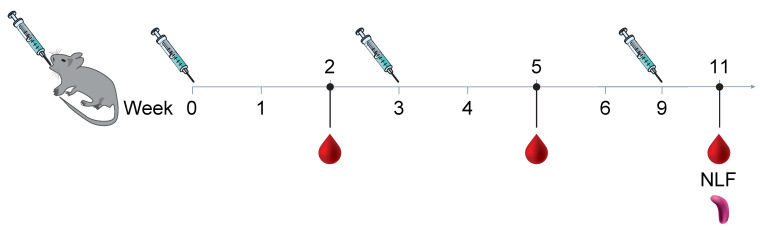
Animals were all immunized three times, three weeks apart using 2 µg micrograms of protein Pre F with aluminum phosphate wet gel suspension (AdjuPhos^®^), inactivated spores (iSpores), live *B. subtilis* spores (spores), or a combination of spores and AdjuPhos^®^. The vaccine was delivered in a total volume of 0.01 mL intranasally (IN) under anesthesia, 0.1 mL (0.05 mL per leg) intramuscularly (IM) while alert, or 0.5 mL orally (PO) while alert.

**Figure 2 biomedicines-13-01112-f002:**
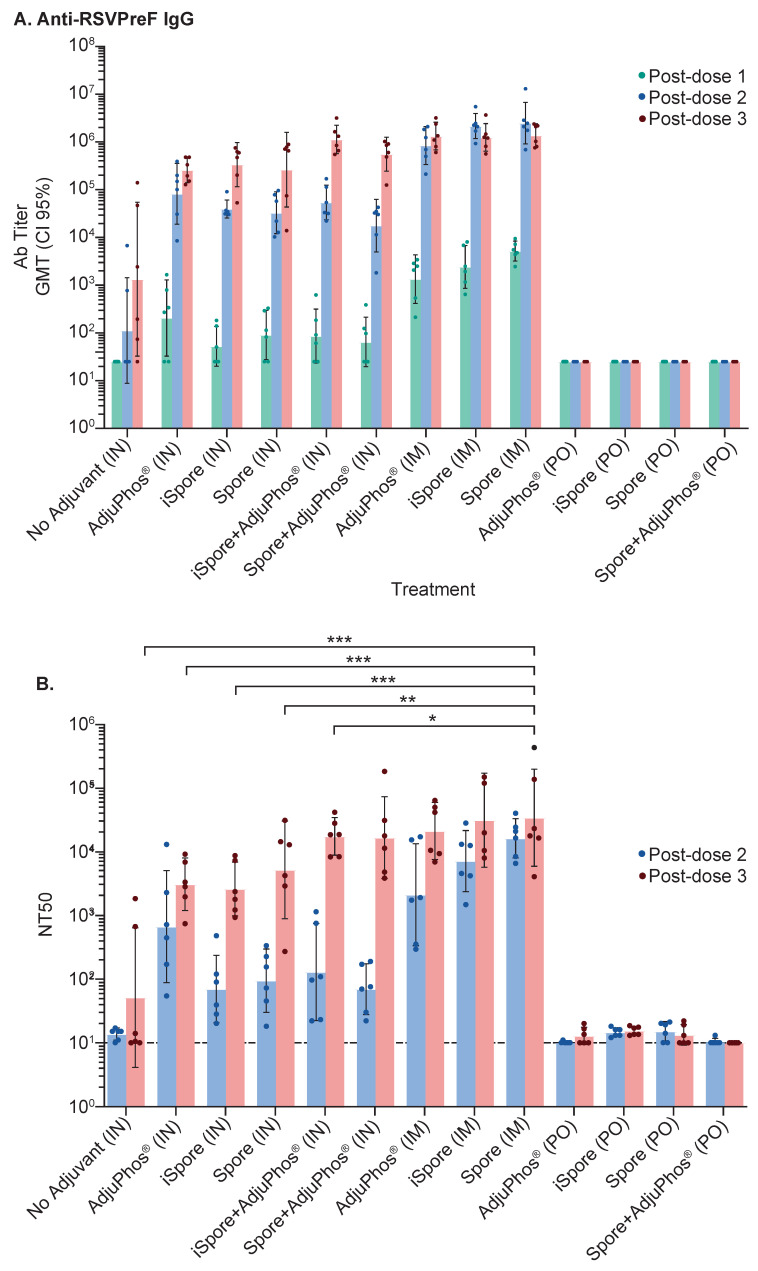
Immune Responses in Mice. Blood collected two weeks after each immunization at the study’s conclusion was used to evaluate immune responses. (**A**) Binding titers against the Pre F protein were assessed using ELISA. The highest reciprocal dilution with an OD reading above the determined background was used as the antibody titer and shown as a geometric mean (GMT) with confidence intervals set at 95%. (**B**) Serum-neutralizing titers (NT50) were measured as the amount of polyclonal serum (reciprocal dilution) needed to reduce 50% of the virus. Both post-dose 2 (PD2) and 3 (PD3) are represented to illustrate the boost from the second to the third immunization. Using two-way ANOVA and Šídák’s multiple comparisons between all the IN groups (except for the group with live spores and aluminum) and the group immunized with F protein plus live spores show significant differences: no adjuvant (IN) vs. spore (IM), *p* value = 0.0004; AdjuPhos^®^ (IN) vs. spore (IM), *p* value = 0.0009; iSpore (IN) vs. spore (IM), *p* value = 0.0008; spore (IN) vs. spore (IM), *p* value = 0.003; iSpore + AdjuPhos^®^ (IN) vs. spore (IM), *p* value = 0.0176. * *p* ≤ 0.05, ** *p* ≤ 0.01, *** *p* ≤ 0.001.

**Figure 3 biomedicines-13-01112-f003:**
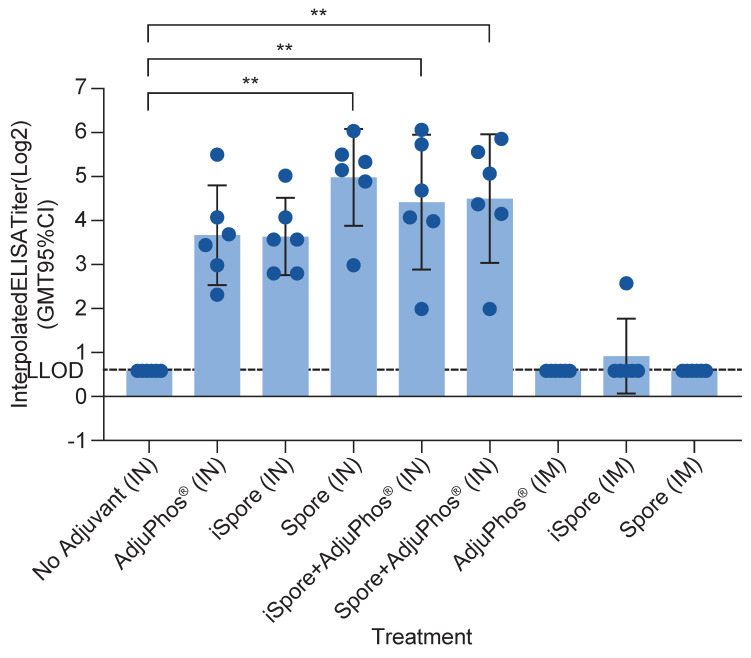
IgA antibody in NAL Samples in Mice. IgA was determined using nasal wash fluid for each group. Bonferroni’s multiple comparisons test shows a significant difference between the following groups: no adjuvant (IN) vs. iSpore (IN), *p* value = 0.0003; no adjuvant (IN) vs. iSpore + AdjuPhos^®^ (IN), *p* value = 0.0052; no adjuvant (IN) vs. spore + AdjuPhos^®^ (IN), *p* value = 0.0055. ** *p* ≤ 0.01.

**Figure 4 biomedicines-13-01112-f004:**
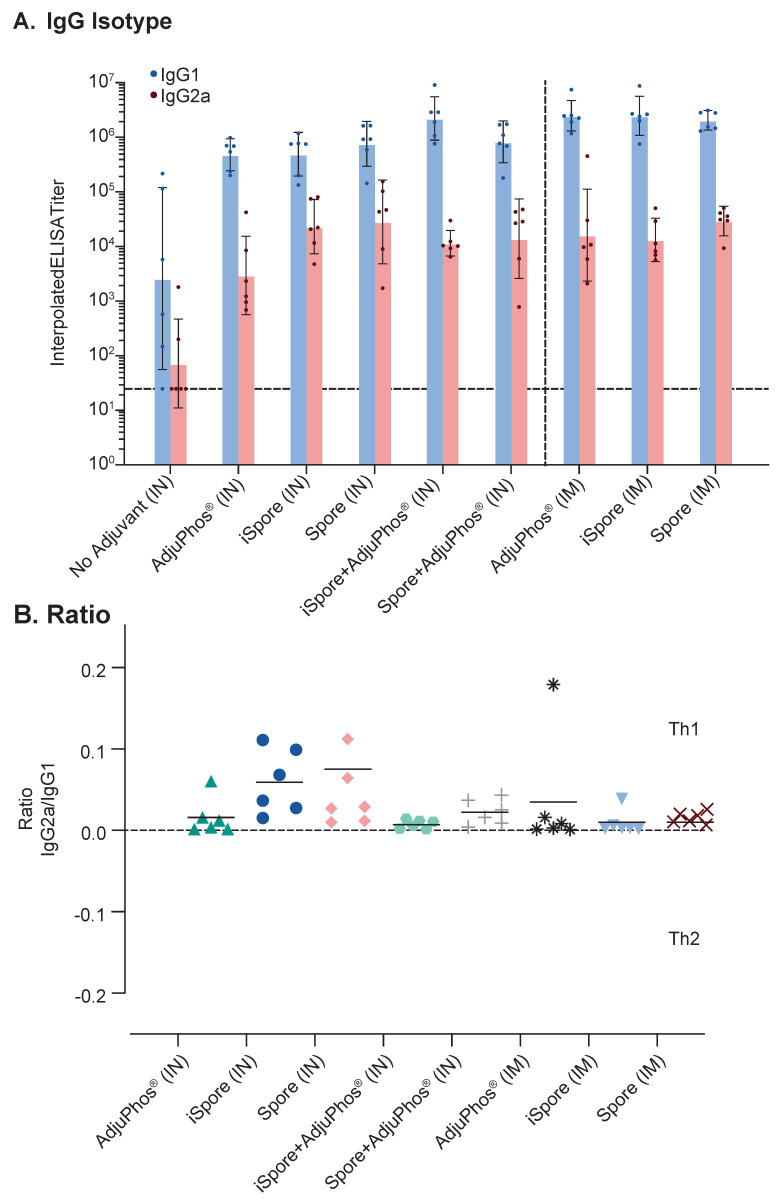
Comparison of Immunoglobulin Isotypes and Ratio in Mice. Balb/c mice with various formulations of RSV Pre F show (**A**) magnitude difference between IgG1 (blue) and IgG2a (orange) and (**B**) overall low IgG2a/IgG1 ratios. Pre F was administered at 2 mcg protein. All intranasal (IN) immunizations were administered at a volume of 0.01 mL. No group was significantly different compared to the group receiving protein with AdjuPhos^®^ adjuvant using Tukey’s multiple comparison and one-way ANOVA tests. All symbols in Figure 4B represent different formulations used for differentiating the groups.

**Figure 5 biomedicines-13-01112-f005:**
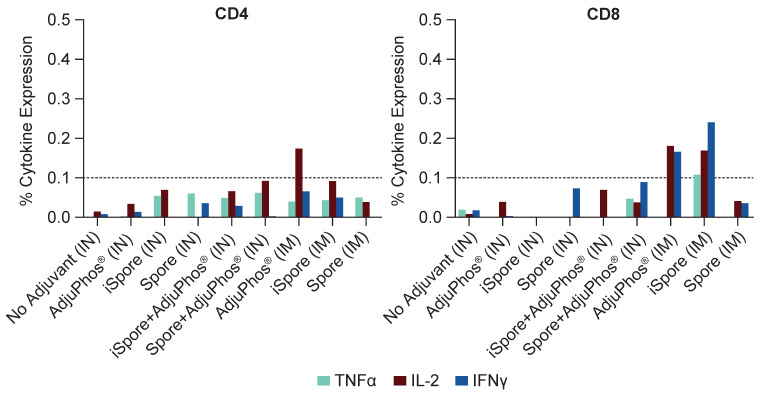
T Cell Responses in Mice. T cell ICS assay was performed by stimulation using pooled mouse splenocyte samples with overlapping peptide pools at 2 mcg per milliliter from the RSV F protein, DMSO, or PMA/Ionomycin. Flow cytometry was utilized to measure T cell expression of IFNγ, TNFα, and IL-2. Overall, CD4 and CD8 T cell responses were low or below the limit of positivity for most experimental groups, with one exception: IL-2 in the aluminum-only group, IM. CD8 T cell responses for IM RSV F and inactivated spore (iSpore) induced similar CD8 T cell responses to RSV F + aluminum, IM.

**Figure 6 biomedicines-13-01112-f006:**
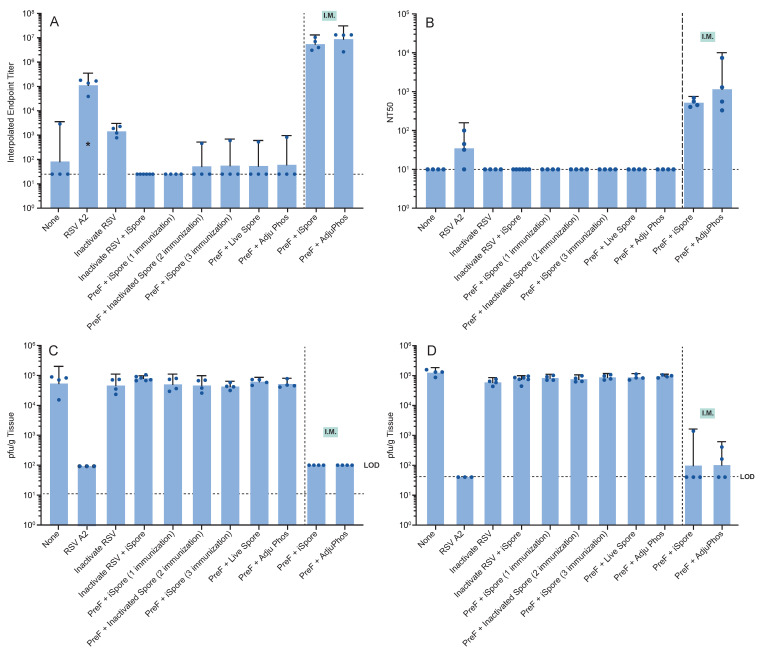
Immune Responses and Viral Load in Cotton Rats Following Strict Intranasal Immunization. Just prior to infection, serum was collected from each animal for antibody determinations. (**A**) Binding antibody was determined via ELISA. (**B**) Serum-neutralizing titers (NT50) were determined from the same blood sample. Four days after infection of the animals with RSV A2, (**C**) lung and (**D**) nose homogenates were evaluated for viral load in each sample type.

**Figure 7 biomedicines-13-01112-f007:**
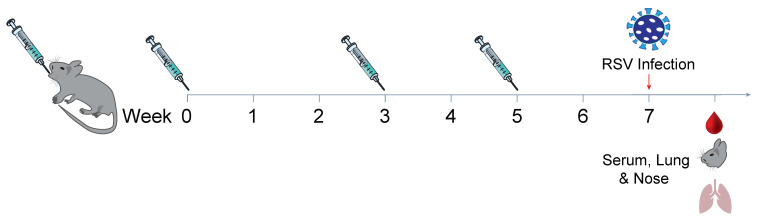
Cotton rats were all immunized intranasally using strict intranasal or intramuscular immunization methods while alert at a volume of 0.1 mL (0.05 mL per leg). Groups were immunized three times except as indicated below. Strict intranasal immunization consisted of 0.01 mL volume delivered to the nostrils of each animal anesthetized with ketamine (40–90 µg per kg) and xylazine (5–10 mg per kg). The groups consisted of no vaccine; live RSV A2 virus; 0.2 mcg F protein-equivalent inactivated RSV with or without iSpores; 2 mcg Pre F + iSpores administered one, two, or three times; and Pre F with live spores or aluminum. In addition, IM groups using 2 mcg of Pre F with iSpores or aluminum were included. Seven weeks after the initial immunization, animals under isoflurane anesthesia were infected with RSV A2. Lung and nose tissues were obtained four days post-infection.

**Figure 8 biomedicines-13-01112-f008:**
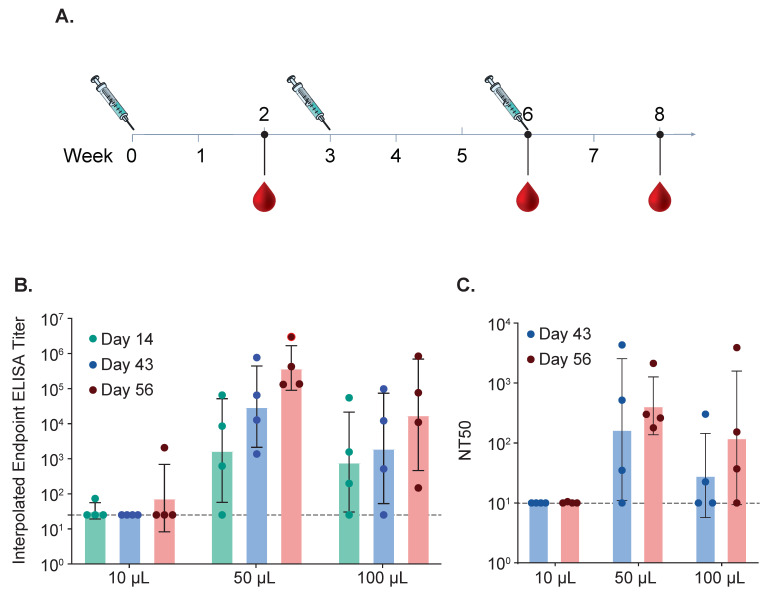
Immune Responses in Cotton Rats After Strict and Non-Strict Intranasal Immunization. (**A**) Cotton rats were immunized intranasally using strict or non-strict intranasal immunization methods. Animals were immunized three times. Strict intranasal immunization consisted of 0.01 mL delivered to the nostrils of each animal anesthetized with ketamine (50–100 µg per kg) and xylazine (2–5 mg per kg). Two other groups consisting of either 50 μL or 100 μL were included. Blood for serum was collected after each immunization and binding antibodies using ELISA (**B**) and neutralizing titers (**C**) were determined for each time point for all three groups.

**Table 1 biomedicines-13-01112-t001:** Treatment groups: mouse study.

Group	Protein	Adjuvant	ROA
1	RSV Pre F (2 mcg)	None	IN
2	RSV Pre F (2 mcg)	AdjuPhos	IN
3	RSV Pre F (2 mcg)	Inactivated spore (iSpore)	IN
4	RSV Pre F (2 mcg)	Live Spore	IN
5	RSV Pre F (2 mcg)	Inactivated spore (iSpore) and AdjuPhos	IN
6	RSV Pre F (2 mcg)	Live spore and AdjuPhos	IN
7	RSV Pre F (2 mcg)	AdjuPhos	IM
8	RSV Pre F (2 mcg)	Inactivated spore (iSpore)	IM
9	RSV Pre F (2 mcg)	Live Spore	IM
10	RSV Pre F (2 mcg)	AdjuPhos	PO
11	RSV Pre F (2 mcg)	Inactivated spore (iSpore)	PO
12	RSV Pre F (2 mcg)	Live Spore	PO
13	RSV Pre F (2 mcg)	Live spore and AdjuPhos	PO

Route of administration (ROA), intranasal immunization (IN), intramuscular immunization (IM), by mouth (PO).

**Table 2 biomedicines-13-01112-t002:** Antibody responses for mouse study.

Group	Adjuvant	ROA	IgG Binding ELISA (GMT(95% CI = U, L))	SNA	IgA	Mean IgG2a/IgG1
1	None	IN	1 × 10^3^ [6 × 10^4^, 3 × 10]	5 × 10 [6 × 10^2^, 4]	2	
2	AdjuPhos	IN	3 × 10^5^ [5 × 10^5^, 1 × 10^5^]	3 × 10^3^ [8 × 10^3^, 1 × 10^3^]	13	0.0158
3	Inactivated spore (iSpore)	IN	3 × 10^5^ [1 × 10^6^, 1 × 10^5^]	3 × 10^3^ [7 × 10^3^, 1 × 10^3^]	13	0.0593
4	Live Spore	IN	3 × 10^5^ [2 × 10^6^, 4 × 10^4^]	5 × 10^3^ [3 × 10^4^, 9 × 10^2^]	32	0.0752
5	Inactivated spore (iSpore) and AdjuPhos	IN	1 × 10^6^ (2 × 10^6^, 6 × 10^5^)	2 × 10^4^ (3 × 10^4^, 9 × 10^3^)	22	0.0073
6	Live spore and AdjuPhos	IN	6 × 10^5^ [1 × 10^6^, 2 × 10^5^]	2 × 10^4^ [7 × 10^4^, 4 × 10^3^]	23	0.0223
7	AdjuPhos	IM	1 × 10^6^ [3 × 10^6^, 7 × 10^5^]	2 × 10^4^ [6 × 10^4^, 7 × 10^3^]	2	0.0348
8	Inactivated spore (iSpore)	IM	1 × 10^6^ [2 × 10^6^, 6 × 10^5^]	3 × 10^4^ [2 × 10^5^, 6 ×10^3^]	2	0.0097
9	Live Spore	IM	1 × 10^6^ [2 × 10^6^, 8 × 10^5^]	3 × 10^4^ [2 × 10^5^, 6 × 10^3^]	2	0.0157
10	AdjuPhos	PO	3 × 10 [3 × 10, 30]	1 × 10 [2 × 10, 9]	ND	ND
11	Inactivated spore (iSpore)	PO	3 × 10 [3 × 10, 30]	2 × 10 [20, 10]	ND	ND
12	Live Spore	PO	3 × 10 [30, 30]	1 × 10 [20, 9]	ND	ND
13	Live spore and AdjuPhos	PO	3 × 10 [30, 30]	1 × 10 [10, 10]	ND	ND

All vaccines contain 2 mcg of RSV Pre F protein. Route of administration (ROA), intranasal immunization (IN), intramuscular immunization (IM), by mouth (PO), geometric mean (GMT), 95% confidence interval (CI) for upper and lower CI (95% CI [U, L).

**Table 3 biomedicines-13-01112-t003:** Treatment groups for cotton rat study 1.

Group		Adjuvant	ROA
1	None	None	-
2	RSV A2 10^5.5^ pfu	None	IN
3	Inactivated RSV (0.2 ug equivalent)	None	IN
4	Inactivated RSV (0.2 ug equivalent)	iSpores 1 × 10^8^ CFU	IN
5	RSV Pre F (2 mcg)	iSpores 1 × 10^8^ CFU	IN
6	RSV Pre F (2 mcg)	iSpores 1 × 10^8^ CFU	IN
7	RSV Pre F (2 mcg)	iSpores 1 × 10^8^ CFU	IN
8	RSV Pre F (2 mcg)	Live spores 1 × 10^8^ CFU	IN
9	RSV Pre F (2 mcg)	AdjuPhos	IN
10	RSV Pre F (2 mcg)	iSpores 1 × 10^8^ CFU	IM
11	RSV Pre F (2 mcg)	AdjuPhos	IM

Route of administration (ROA), intranasal immunization (IN), intramuscular immunization (IM). All IN was delivered at a volume of 0.01 mL. All IM was delivered at a volume of 0.1 mL, equally divided between two legs. Groups 2 and 5 were dosed once; Group 6 was dosed twice; Group 7 was dosed three times.

**Table 4 biomedicines-13-01112-t004:** Antibody and viral titers in respiratory tissues in cotton rat study 1.

Grp	Vaccine	Adjuvant	ROA	IgG Binding ELISA (GMT)	SNA	Lung Virus Reduction Log10	Nose Virus Reduction Log10
1	None	None	NA	80	10	-	-
2	RSV A2 10^5.5^ pfu	None	IN	1 × 10^5^	30	2.6	3.3
3	Inactivated RSV (0.2 ug equivalent)	None	IN	1 × 10^3^	10	>0.1	>0.2
4	Inactivated RSV (0.2 ug equivalent)	iSpores 1 × 10^8^ CFU	IN	30	10	0	>0.2
5	RSV Pre F (2 mcg)	iSpores 1 × 10^8^ CFU	IN	30	10	0	>0.2
6	RSV Pre F (2 mcg)	iSpores 1 × 10^8^ CFU	IN	30	10	0	>0.2
7	RSV Pre F (2 mcg)	iSpores 1 × 10^8^ CFU	IN	50	10	0	>0.2
8	RSV Pre F (2 mcg)	Live spores 1 × 10^8^ CFU	IN	50	10	>0.1	>0.2
9	RSV Pre F (2 mcg)	AdjuPhos	IN	50	10	0	>0.2
10	RSV Pre F (2 mcg)	iSpores 1 × 10^8^ CFU	IM	5 × 10^6^	5 × 10^2^	2.6	3
11	RSV Pre F (2 mcg)	AdjuPhos	IM	9 × 10^6^	1 × 10^3^	2.6	3

Route of administration (ROA), intranasal immunization (IN), intramuscular immunization (IM).

**Table 5 biomedicines-13-01112-t005:** Antibody responses for cotton rat study 2.

Vaccine Volume	IgG Binding ELISA (GMT; [95% CI = U, L])	SNA
0.01	8 × 10 [3 × 10^3^, 2]	10
0.05	4 × 10^5^ [4 × 10^6^, 4 ×10^4^]	4 × 10^2^
0.1	2 × 10^4^ [6 × 10^6^, 50]	1 × 10^2^

## Data Availability

All data are maintained at Merck & Co., Inc., Rahway, NJ, USA, and requests may be made to the corresponding author.

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
