# Peer review of "Immunogenicity of RSV Fusion Protein Adsorbed to Non-Pathogenic Bacillus subtilis Spores: Implications for Mucosal Vaccine Delivery in Nonclinical Animal Models"

_biomedicines, 2025, doi:10.3390/biomedicines13051112_

Round 1
Reviewer 1 Report
Comments and Suggestions for Authors
Your paper puts forward an innovative method for developing mucosal vaccine, and explores the immunogenicity of the fusion protein of RSV and Bacillus subtilis spores through different routes of administration and animal models. This study provides valuable insights into the potential of this platform, and useful information in preclinical research. However, several aspects of the manuscript need to be revised to improve overall quality. The following are specific suggestions to guide your revision:
- Small sample size and limited statistical details are issues of concern. In animal research, small sample size and limited statistical details will undermine the reliability of research results. In order to solve these problems, the widely accepted scientific standard is to conduct at least two animal experiments.
- Comparing Bacillus subtilis spores with established mucosal adjuvants will strengthen this study. If possible, add a control group (such as cholera toxin B subunit). Alternatively, the comparison based on literature can be discussed in Section 4.
- Revise section 5 conclusion, summarize the key findings (for example, the efficacy of mice and the failure of cotton mice).
- Discuss the difference of species and expand the influence on human translation. The different reactions between mice (strong immunity) and cotton mice (no response when strictly administered) indicate species-specific factors. Consider adding a section in Section 4, citing relevant literature, and comparing the nasal anatomy and immune response of these species. Also, the correlation with human vaccine can be discussed in section 4, considering nasal anatomy and immune differences.
- The study will benefit from a deeper understanding of how spores of Bacillus subtilis enhance immunity. Consider adding a speculative mechanism (such as innate immune activation) in Section 4, and propose future experiments (such as cytokine analysis).
Author Response
Please see the attachment. Thank you for your comments and your commitment to the peer review process.

Reviewer 2 Report
Comments and Suggestions for Authors
The manuscript presents a well-structured, original and thoroughly conducted study investigating the immunogenicity of an RSV vaccine candidate using Bacillus subtilis spores as a delivery platform. The study is scientifically sound and relevant to the current need for mucosal vaccines. The study addresses important variables such as animal models, strict intranasal versus non-strict administration and differences between systemic and mucosal responses. However, there are some concerns that need to be addressed before publication
Abstract
- Please state the aims of your study clearly.
Introduction
- Provide a more detailed context to the RCV.
- L41-46: This statement fits better with the “Results” section.
- You could explain in detail why cotton rats are an appropriate model.
Materials and Methods
- Please give details of the “sacrifice” you refer to in L76.
- Explain the different groups and the animals included in these groups. It is somewhat confusing for the reader to understand how many groups and how many animals were included in the trial.
- Was a sample size calculation performed?
Results
- Consider summarizing key findings visually. In particular, a chart or table summarizing the immune outcomes by species, route and method of administration would help to identify contrasts at a glance.
Discussion:
- Some points are repeated several times (e.g., failure of oral administration, efficacy of spores in mice but not in rats). Streamlining would improve the flow.
- A sentence could be added to emphasize that subtilis remains promising, but that optimization of administration and formulation is essential.
Minor comments
- Moderate editing regarding English language is required for the improvement of the text.
Author Response

(The authors gave the same response as above.)

Round 2
Reviewer 1 Report
Comments and Suggestions for Authors
Dear authors:
Thank you for your comprehensive revision and detailed reply to my comments. Your explanation of sample size rationality, species-specific differences and mechanical hypothesis is satisfactory, and the concorns raised are addressed. The revision of the discussion and conclusion properly links the findings and limitations of the study. In view of your clear reply and improvement of the manuscript, I support publishing your work in its current form. This study provides valuable insights for this field.
Author Response
Our sincere gratitude for your commitment to the peer review process and for the opportunity to share our work. We appreciate your insights and constructive feedback to enhance how our work is shared.
Reviewer 2 Report
Comments and Suggestions for Authors
I have carefully reviewed the revised manuscript and the authors' responses to my previous comments. I am pleased to report that the authors have adequately addressed all the points I raised previously. They have provided clear explanations, made the necessary revisions to the manuscript, and improved both the clarity and overall quality of the paper.
I consider the manuscript in its current form to be significantly improved and ready for publication.
Recommendation: Accept
Author Response
We like to share our sincere gratitude for your commitment to the peer review process and for the opportunity to share our work. We appreciate your constructive feedback that we believe enhances how our work will be shared. Thank you.